# Community-Level Analysis of Drinking Water Data Highlights the Importance of Drinking Water Metrics for the State, Federal Environmental Health Justice Priorities in the United States

**DOI:** 10.3390/ijerph181910401

**Published:** 2021-10-02

**Authors:** Uloma Igara Uche, Sydney Evans, Soren Rundquist, Chris Campbell, Olga V. Naidenko

**Affiliations:** 1Environmental Working Group, 1250 I Street NW, Suite 1000, Washington, DC 20005, USA; chris@ewg.org (C.C.); olga@ewg.org (O.V.N.); 2Environmental Working Group, 111 Third Avenue South, Suite 240, Minneapolis, MN 55401, USA; srundquist@ewg.org

**Keywords:** drinking water, cumulative cancer risk, environmental health, environmental justice

## Abstract

Research studies analyzing the geospatial distribution of air pollution and other types of environmental contamination documented the persistence of environmental health disparities between communities. Due to the shortage of publicly available data, only limited research has been published on the geospatial distribution of drinking water pollution. Here we present a framework for the joint consideration of community-level drinking water data and demographic data. Our analysis builds on a comprehensive data set of drinking water contaminant occurrence for the United States for 2014–2019 and the American Community Survey 5-year estimates (2015–2019) from the U.S. Census Bureau. Focusing on the U.S. states of California and Texas for which geospatial data on community water system service boundaries are publicly available, we examine cumulative cancer risk for water served by community water systems of different sizes relative to demographic characteristics for the populations served by these water systems. In both California and Texas, greater cumulative cancer risk was observed for water systems serving communities with a higher percentage of Hispanic/Latino and Black/African American community members. This investigation demonstrates that it is both practical and essential to incorporate and expand the drinking water data metrics in the analysis of environmental pollution and environmental health. The framework presented here can support the development of public policies to advance environmental health justice priorities on state and federal levels in the U.S.

## 1. Introduction

Geospatial analysis of environmental pollution data can help policymakers and communities develop public health-protective policies such as control of pollution sources at the local level and prioritization of infrastructure investment. Localized, geographically targeted data sets on air pollution, now available in many countries, revealed inequalities in exposures to air contaminants among different communities and populations [1,2]. In the United States, disparities in community exposure to air pollution were observed and reported in studies conducted both on national and local levels [3,4]. There is also growing evidence of social disparities in drinking water quality across the U.S., with community water systems that serve areas with a greater percentage of people of color being more likely to experience worse drinking water quality compared to nationwide averages [5,6,7]. However, research on the geospatial distribution of drinking water contamination has been limited by the lack of comprehensive, centralized data sets on water use, water quality, and water service areas [8]. In the U.S., California pioneered geospatial analysis of drinking water data under the CalEnviroScreen program [9,10], yet detailed analysis of geospatial patterns in drinking water quality for other U.S. states has not been published.

Nearly 90% of the U.S. population receives their drinking water from community water systems that supply water to the same population year-round. Communities and households that do not have public water services typically use private wells. For both drinking water systems and private wells, source water contamination and tap water quality are a constant, significant concern. A recent workshop report from the U.S. National Academies of Sciences, Engineering, and Medicine noted that U.S. drinking water is at risk from contamination due to pollution of water sources, aging infrastructure for water treatment and water distribution, and extreme weather patterns associated with climate change, such as droughts, wildfires, and flooding [11].

Here, we present case studies for the states of California and Texas evaluating drinking water quality data for community water systems and demographic data available from the U.S. Census Bureau. For this analysis, drinking water quality was assessed in terms of cumulative cancer risk due to the presence of multiple carcinogenic contaminants [12,13]. The overall goal of this research is to develop a framework that can identify and describe patterns in drinking water quality on the community level. Such analysis can help state and federal efforts in the U.S. to improve drinking water quality for all communities and to advance environmental health justice priorities.

## 2. Materials and Methods

To conduct this analysis, our research team assembled a drinking water data set for the United States, compiling results of drinking water testing that community water systems conducted from 2014 to 2019 to demonstrate compliance with drinking water quality regulations. The combined nationwide data set includes drinking water test results for all 50 U.S. states, with a total of 47,820 community water systems. The underlying data sets for each U.S. state can be obtained either via a direct data access portal provided by a state drinking water authority or via a records request. Appendix A, Table A1 lists the links for the websites of government agencies from which data were obtained.

Additionally, we retrieved results of drinking water tests for unregulated contaminants (substances that may be present in drinking water but that have no legal limits under the current U.S. regulations) from the U.S. Environmental Protection Agency’s fourth Unregulated Contaminant Monitoring Rule (UCMR4) assessment monitoring [14]. Data for the group of 9 haloacetic acids were downloaded from the U.S. EPA website in July 2021, corresponding to testing conducted in 2018–2021. The compiled data set with data from both sources is viewable online on the website of our organization at https://www.ewg.org/tapwater/, accessed on 25 June 2021.

To define contaminant concentration values for each community water system, all available test results for each system for the study period were included in the calculation of arithmetic means. Test results reported in the original state data sets as “non-detects” were assigned a value of zero and included in the overall data array for the calculation of arithmetic averages. Calculations of estimated cumulative cancer risk were performed as described in previous studies [12,13]. The cumulative cancer risk due to all contaminants at a system level was calculated using the following equation:$$R\;=\sum_{i=1}^{N}\frac{C_{i}}{B_{i}}$$
where:

i is the individual contaminant,

N is the total number of contaminants,

R is the cumulative lifetime cancer risk per million people on a community water system level due to all drinking water contaminants,

C_i_ is the long-term average contaminant concentration in a community water system, calculated as an arithmetic average of all test results for the specified period,

B_i_ is the cancer risk benchmark that represents the contaminant concentrations corresponding to 10^−6^ lifetime cancer risk, and

Σ denotes the summation of all contaminants in a community water system.

Appendix A, Table A2 lists all U.S. EPA maximum contaminant levels (MCLs) and cancer risk benchmarks used for the calculations of risk estimates presented in this article. For the state-specific case studies, we used the publicly available GIS data layers for water system service area boundaries for California (https://gispublic.waterboards.ca.gov/portal/apps/webappviewer/index.html?id=272351aa7db14435989647a86e6d3ad8 (accessed on 30 March 2021)) and Texas (https://www3.twdb.texas.gov/apps/WaterServiceBoundaries/), accessed on 30 March 2021. To characterize the demographic of the population served by community water systems, tract-level demographic data were obtained from the U.S. Census Bureau’s American Community Survey 5-year estimates for the years 2015–2019 (Available at https://acsdatacommunity.prb.org/discussion-forum/f/forum/641/u-s-census-bureau-releases-2015-2019-acs-5-year-estimates, accessed on 9 July 2021). According to the U.S. Census Bureau, census tracts generally have a population size between 1200 and 8000 people.

Using GIS software ArcMap 10.8 (ESRI Inc., Redlands, CA, USA), census tract boundaries were matched to community water system boundaries, and the percentage of overlap by area was estimated. Using the percentage of overlap between the census tract and each community water system, tract-level demographic characteristics were proportionally assigned to each community water system and then summed for each system. An illustrative example presents a calculation of demographic variable Y for the population served by a community water system (Y_CWS_) where the water system service area covers x%, y%, and z% of tracts A, B, and C by area, respectively, with the following equation:(Y_CWS_) = [(Y_A_ × x%) + (Y_B_ × y%) + (Y_C_ × z%)]
where Y_A_, Y_B,_ and Y_C_ are the reported frequency of the demographic variable Y within each census tract that partially or completely overlaps with the service area for the community water system.

The data set for census tracts within community water systems was linked to the contaminant occurrence data set using public water system identification numbers (PWSID), unique identifiers for each community water system in the U.S. To be included in the analysis, the systems had to be represented in both the GIS service area data set and the national contaminant occurrence data set. Data were processed and visualized using STATA software version 17.0 (StataCorp, College Station, TX, USA) and analyzed using SAS software version 9.4 (SAS Institute Inc., Cary, NC, USA).

## 3. Results

### 3.1. Combined Data Set for Drinking Water Contaminant Occurrence for the U.S.

The drinking water data set analyzed here included data for 47,820 community water systems across the United States. Jointly, these water systems serve a population of approximately 306 million people. In United States drinking water, the most common contaminants are arsenic, nitrate, radium, and disinfection byproducts (Table 1). In the disinfection byproducts category, national data are available for the group of four trihalomethanes (abbreviated as THM4) that includes chloroform, bromoform, dibromochloromethane, and bromodichloromethane; the group of five haloacetic acids (HAA5) that includes monochloroacetic acid, dichloroacetic acid, monobromoacetic acid, dibromoacetic acid, and trichloracetic acid; and the group of nine haloacetic acids (abbreviated as HAA9), which includes the HAA5 group as well as bromochloroacetic acid, bromodichloroacetic acid, chlorodibromoacetic acid, and tribromoacetic acid [13,15].

We note that Table 1 and this study overall include data for contaminants for which recent U.S.-wide testing data are available. Recent studies suggested that other contaminants, especially per- and polyfluoroalkyl substances, or PFAS, may be widely present in drinking water for millions of people [16]. However, due to the lack of nationwide testing data for the United States, PFAS could not be included in the present study. The U.S. EPA maximum contaminant levels, or MCLs, are the legal standards for community water systems established under the U.S. Safe Drinking Water Act (Appendix A, Table A2). The average concentrations of contaminants (Table 1) are lower than the legal limits established by the U.S. EPA (Table A2). However, it is important to recognize that numerous community water systems in the U.S., serving millions of people, are not able to comply with the existing standards and have average contaminant concentrations that are close to or violate the existing legal standards [17].

The UCMR4 data set for the occurrence of the group of nine haloacetic acids (HAA9) includes data for 4733 community water systems, serving a combined population of approximately 251 million people (Table 1), around 82% of the approximately 306 million people in the United States that are served by community water systems. Figure 1 shows the concentration of HAA9 among community water systems with data available under the UCMR4 program (data accessed from the U.S. EPA website in July 2021).

Calculations of estimated cumulative cancer risk for co-occurring drinking water contaminants were performed following an approach developed in the U.S. EPA’s Air Toxics Assessment [18], as described in previous studies [12,13]. In this methodology, a cancer risk estimate indicates the probability of getting cancer over a lifetime of exposure to a mixture of carcinogenic contaminants present at specific concentrations. Analysis of cumulative cancer risk for the water systems included in this data set demonstrates that estimated cumulative cancer risk varies by system size (Figure 2). Large and medium-sized water systems typically depend on surface water for all or part of their water supply and generally have higher levels of disinfection byproducts. In contrast, systems that rely on groundwater for their supply tend to have higher levels of nitrate and arsenic in both source and treated water. Figure 2 shows the overall cumulative cancer risk due to all contaminants found in community water systems serving populations of different sizes.

For the calculation of cumulative cancer risk shown in Figure 2, data for nine haloacetic acids (HAA9) were used where available (4733 systems). For other systems in the overall data set, only data for five regulated haloacetic acids (HAA5) are available. Where both HAA5 and HAA9 concentrations were available for a water system, cancer risk calculations used the HAA9 value. Partial availability of HAA9 concentrations underestimates the cumulative cancer risk for some systems.

Further, data in Figure 2 demonstrate that system size is a factor relevant to the contaminant occurrence profile and cumulative cancer risk. Community size is also a factor relevant to the economics of operating a water system and installation of contaminant-removal technologies since small- and medium-sized systems often lack sufficient resources necessary for infrastructure upgrades. Within the U.S. EPA UCMR4 data set for the HAA9 contaminant group, all or nearly all community water systems serving populations of 10,000 people or more are included; however, only a limited number of systems serving populations less than 10,000 currently have HAA9 data.

### 3.2. Attribution of Demographic Data to Community Water Systems

State agencies and individual researchers across the U.S. have started the process of geocoding community water system boundaries. Such geocoded data sets, once publicly available, would greatly facilitate future research in this field. For this study, we focused on the U.S. states of California and Texas. These two states are the most populous within the U.S., with an estimated 2021 population of 39 and 29 million people, respectively. The GIS boundaries for community water systems in these two states are posted on state websites (links for the data sets listed in the Methods section). We note that, since the development of community water system boundary data sets is an ongoing process, the data used here might not include all community water systems in a state and/or exact information on the service boundaries of a specific system.

To create a combined data set for each state with demographic and contaminant occurrence data for community water systems, water systems were matched across service area and water quality data sets by their public water system identification (PWSID) numbers. We note that this study did not include an analysis of populations that depend on private wells as a source of their drinking water. Our analysis also did not include two categories of systems: (1) non-community water systems (systems that serve transient or non-transient populations that do not depend on the water provided by the system year-round) and (2) wholesale systems (identified by a reported served population of zero), which are systems that produce treated, finished water and then deliver some or all of that finished water to another public water system. The matched data sets with both water quality and service area boundary data included 2721 community water systems for California and 4364 water systems for Texas (Figure 3).

To characterize the demographic characteristics of the population served by systems, tract-level demographic data from the American Community Survey data set for 2015–2019 were proportionally assigned and then summed for each community water system in the data set, as described in the methodology. This approach assumes that the population is evenly distributed within the tract, an assumption that, while necessary for the purposes of this methodology, presents a simplified view of population distribution within each tract, which may be heterogeneous.

Figure 4 shows the correlation between the population values for community water systems that are (a) calculated based on census tracts that are wholly or partially included within the boundaries of a community water system and (b) estimates of the population served by community water systems according to the U.S. EPA Envirofacts database. Overall, a visual correlation between the two values is observed (Figure 4), indicating the methodology of proportional assignment of census tract data to community water systems results in reliable estimates.

For very small community water systems serving populations of 500 people or less, there was no obvious correlation between population estimates calculated with the tract assignment method versus population estimates reported in the U.S. EPA Envirofacts database (Figure 5). Community water systems of this size tend to be in rural areas with lower population density. Demographic data assignment for community water systems serving 500 people or less would likely require a smaller geographic unit, such as a census block, to accurately match the population count and other demographic characteristics within a community water system service area.

To examine patterns in overall water quality across water system size and demographic data for the populations served, we grouped community water systems based on the size of the served population according to the data from the U.S. EPA Envirofacts. This grouping included very small systems serving populations of 500 people or fewer, small (populations of 500–3300), medium (3301–10,000), large (10,001–100,000), and very large (more than 100,000) (Figure 6). Correlation between the population calculated by the census tract assignment method and the reported population served by the system according to the U.S. EPA Envirofacts database appeared to increase with system size (Figure 4 and Figure 5).

For further analysis in this study, we focused on water systems serving populations of more than 500 people, and for which the tract-assigned population was ±50% of the U.S. EPA-reported population for each community water system. With the tract assignment method, the number of census tracts that are partially or completely included in a community water system boundary is, as anticipated, proportionally correlated with the geographic size of a water system service area (Figure 7). With this final criterion included in our analysis, a total of 2638 systems were analyzed in California, serving an estimated population of 35 million people, and a total of 4297 systems were analyzed for Texas, serving an estimated population of nearly 27 million people.

### 3.3. Analysis of Cumulative Cancer Risk and Demographic Data for Community Water Systems

As a case study for the joint analysis of drinking water data and demographic data, we examined demographic characteristics defined as race and ethnicity within the U.S. Census Bureau data set. As described by the U.S. Census, “*the racial categories included in the census questionnaire generally reflect a social definition of race recognized in [the U.S.] and not an attempt to define race biologically, anthropologically, or genetically*” [19]. For this case study, we used the following demographic groups from the U.S. Census: “Black or African American” and “Hispanic or Latino origin” [19]. The demographic terms used in our study are the terms used by the U.S. Census. Both California and Texas have a large proportion of people who self-identify as belonging to one or both demographic groups. According to the U.S. Census QuickFacts (https://www.census.gov/quickfacts, data accessed on 3 August 2021), 39.4% of the population in California and 39.7% of the population in Texas identified as Hispanic or Latino; and 6.5% of the population in California and 12.9% of the population in Texas identified as Black or African American.

Based on the tract assignment methodology developed here, each system was assigned a percentage value for the percent of the population that is “Black or African American” and percent of the population that is of “Hispanic or Latino origin”. These percentage values are a proportional representation of the Black or African American population and Hispanic or Latino origin population in the census tracts that are partially or wholly encompassed in the community water system service area. For statistical analysis, community water systems were arrayed according to the percentage of the population that is Black/African American or Hispanic/Latino and then divided into terciles. For each tercile, we calculated the median cumulative cancer risk due to multiple contaminants. Then a Jonckheere-Terpstra statistical test was used to examine significant trends in the median cumulative cancer risk relative to the percentage of community members that are Black/African American or Hispanic/Latino. To control for the effects of system size and the differential availability of UCMR4 data, we also analyzed the relationship between cumulative cancer risk and demographic data within subsets of community water systems based on size. Figure 8 shows the distribution of cumulative cancer risk among community water systems arrayed according to the percentage of the population that is Black/African American or Hispanic/Latino within water system service areas.

As documented in Figure 9, increased cumulative cancer risk is observed in the terciles of community water systems that serve a larger proportion of either Black/African American community members or Hispanic/Latino community members. These results are statistically significant, with trend test *p* values smaller than 0.05.

We repeated the tercile cumulative risk analysis within community water systems arrayed by system size into four groups: systems serving populations of 501–3300 people (defined by the U.S. EPA as small community water systems); systems serving populations of 3301–10,000 people, defined as medium community water systems; large systems serving 10,001–100,000 people; and very large systems serving populations greater than 100,000 people (Table 2 and Table 3). For each group of community water systems, terciles were defined according to the percentage of the population that is Black/African American or Hispanic/Latino within community water system service areas. With rounding, each tercile has the same number of community water systems. Due to differences in the size of the population served by individual community water systems, the total population served differs between terciles.

For the state of California, analysis of cumulative cancer risk for community water systems grouped by system size identified a statistically significant trend for worse water quality in very large community water systems that have a greater percentage of Black or African American residents (Table 2). In small community water systems in California, statistically significant greater cumulative cancer risk was observed for systems with a greater percentage of Hispanic or Latino community residents. Even when statistically significant trends are not observed (trend test *p* values larger than 0.05), it is striking that median cumulative cancer risk is greater in Tercile 3 (T3) with the greater percentage of Black or African American residents compared to the median risk for Tercile 1 (T1) and Tercile 2 (T2) in medium and large community water systems.

For the state of Texas, the striking finding highlighting environmental health injustice is the greater cumulative cancer risk for community water systems that serve a larger percentage of Hispanic or Latino residents (Table 3). These findings show a statistically significant trend of higher cumulative cancer risk for small and medium community water systems serving a larger proportion of Hispanic/Latino community members. This statistically significant trend is not observed for large and very large water systems in Texas; however, even for those systems, the tercile with the highest percentage of Hispanic or Latino residents (T3) has a higher median cumulative cancer risk compared to the tercile with the smallest percentage of Hispanic or Latino residents (T1). Small community water systems in Texas serving a larger proportion of Black or African American community members also had higher cumulative cancer risk.

## 4. Discussion

The development and implementation of national and local policies to advance environmental health justice require data metrics for addressing environmental pollution that integrate data on contaminant exposure from all sources, including air, drinking water, and other media. In the United States, the U.S. Environmental Protection Agency Environmental Justice Screening and Mapping Tool (EJSCREEN) includes a variety of important indices and indicators such as air quality, lead paint, and proximity to various types of industrial facilities and potentially contaminated sites [20]. However, U.S. EPA EJSCREEN does not currently include drinking water data, a significant limitation given the importance of considering drinking water quality in national-level environmental justice policies. A geospatial analysis of drinking water data on a community water system level can provide the necessary information to fill this gap, establishing a more comprehensive approach for environmental health justice policies.

Here we present the summary statistics for a drinking water data set for 50 U.S. states for data years 2014–2019, as well as case studies for California and Texas where we examined the intersection of drinking water quality data and demographic data on a community water system level. For both states analyzed in this study, greater cumulative cancer risk was observed for water systems serving communities with a higher percentage of Hispanic/Latino and Black/African American community members. The results of our analysis are consistent with published literature finding social disparities in drinking water quality. A study conducted in California’s San Joaquin Valley reported higher arsenic concentrations in drinking water and higher odds of having a violation of drinking water standards for systems serving predominantly socio-economically disadvantaged communities [21]. A U.S.-wide study reported a significant association between the percent of Hispanic residents served by a community water system and average concentrations of nitrate in drinking water [22]. A U.S.-wide study conducted by the Natural Resources Defense Council, Environmental Justice Health Alliance for Chemical Policy Reform and Coming Clean found that water systems serving communities of color had a higher rate of violations of national drinking water standards [7].

The findings of our study, together with prior research, highlight how communities of color disproportionately face worse drinking water quality, which can increase their risk for adverse health impacts such as the elevated risk of cancer. These disparities in environmental contaminant exposure are further aggravated by the fact that communities and populations of color in the United States continue to experience greater health inequalities [23] in general. They also have less access to adequate health care compared to other populations [24]. According to the U.S. National Cancer Institute’s Surveillance, Epidemiology, and End Results (SEER) database, cancer mortality rates are higher among the Black/African American population than other groups [25]. Including drinking water data metrics in state and federal environmental justice analyses used for policy decisions and other actions would be an important step to address the identified disparity while also offering better health protections for all communities.

Several aspects of our framework build on prior approaches and methodologies and expand them further. First, the use of a water system service area as a study unit allows for the analysis of potential differences in drinking water quality between communities. Overall, drinking water quality is expected to be consistent across the service area for a specific community water system. A key exception to that assumption is lead contamination from water pipes. The presence of lead service lines in specific parts of a community water system service area would translate into higher concentrations of lead at the taps in homes, schools, and other buildings in those areas. As documented in a report by the U.S. Government Accountability Office, identifying and remedying lead hazards in drinking water requires more precise geospatial data for lead service line locations [26]. Additionally, the concentrations of disinfection byproducts can vary in different service locations within community water system boundaries. Finally, the presence of either multiple drinking water wells in a groundwater-based system or more than one water treatment plant within a water system service area can contribute to water quality differences at the tap within community water system boundaries. Even with these limitations, the use of a community water system service area as a study unit is an important step forward relative to prior studies that use larger geographic units such as “county”, the approach used in the U.S. EPA Environmental Quality Index [27]. These types of units do not always align well with water system service areas and therefore do not always accurately represent the population within the service area.

Mapping studies integrating drinking water metrics with indices for other environmental indicators, especially air quality, have used “census tract” as a study unit, an approach applied in CalEnviroScreen, California’s data framework for evaluating environmental pollution exposure [10]. Depending on the size of a geographic area covered by a community water system and the geographic size of census tracts in the same location, either multiple water systems may overlap within a single census tract, or multiple tracts may be partially or completely included within a water system (Figure 7). This consideration of relative sizes of study areas may determine whether a tract-level or a system-level evaluation may be most appropriate for a specific analysis. As Figure 5 demonstrates, the census tract-based methodology did not produce a suitable fit for community water systems that serve populations of 500 people or fewer. Our study used the census tract data from the American Community Survey 5-year estimates for 2015–2019 published by the U.S. Census Bureau. The anticipated availability of the census block-level results for the 2020 U.S. Census would allow future studies to apply this framework developed by our study with greater precision using the smaller geographic unit of census blocks.

A unique feature of our analysis is the use of a carcinogenic potency-based estimates of cumulative cancer risk due to contaminants present in drinking water. This risk estimate integrates the information about the overall risk from co-occurring carcinogenic contaminants in drinking water, a methodology originally developed for carcinogenic air pollutants and implemented in the U.S. EPA National Air Toxics Assessment [18]. Previous research on drinking water quality focused on the presence of specific contaminants and violations of existing drinking water standards. For communities around the country whose water systems might not be able to meet existing national drinking water standards, the first urgent priority is to access financial and institutional resources to support drinking water infrastructure improvements. However, both research and policy applications should include the data on the occurrence of *all* pollutants that may harm human health, not just contaminants that may be currently regulated on the federal or state levels. New research continues to demonstrate that long-known contaminants affect human health at concentrations significantly lower than existing legal standards [28]. Further, emergent contaminants, such as per- and polyfluoroalkyl substances that currently do not have drinking water standards in the U.S., can harm human health in a variety of ways, including increased risk of cancer [29].

In future studies, cumulative cancer risk due to the presence of drinking water contaminants should be integrated with cumulative cancer risk due to the presence of air pollutants. A combined estimate of cancer risk due to the presence of air and water contaminants on a census tract level, or another spatial unit as appropriate for a specific study, would be informative for future epidemiological research and integrative assessment of environmental quality. Multiple factors such as genetic background and individual life circumstances can influence cancer risk. The assessment of cumulative exposures as well as other risk factors should become a cornerstone for future policies to improve environmental health in all communities, especially communities that historically experienced discrimination and may still face a disproportionate burden of pollution to date. Implementing the data and methods presented here into screening tools can supplement and support community-level knowledge in facing environmental challenges. The framework presented here for the combined analysis of drinking water data and demographic data can support the development of environmental justice policies on state and federal levels in the United States.

## 5. Conclusions

Based on the analyses conducted here, we recommend that government agencies and policymakers consider geospatial, community-level data on drinking water contaminant occurrence in research and policy initiatives that aim to promote environmental health justice. Environmental justice screening and mapping tools should include sociodemographic analysis and data on drinking water contaminants in addition to information about compliance with drinking water health standards and monitoring requirements.

## Figures and Tables

**Figure 1 ijerph-18-10401-f001:**
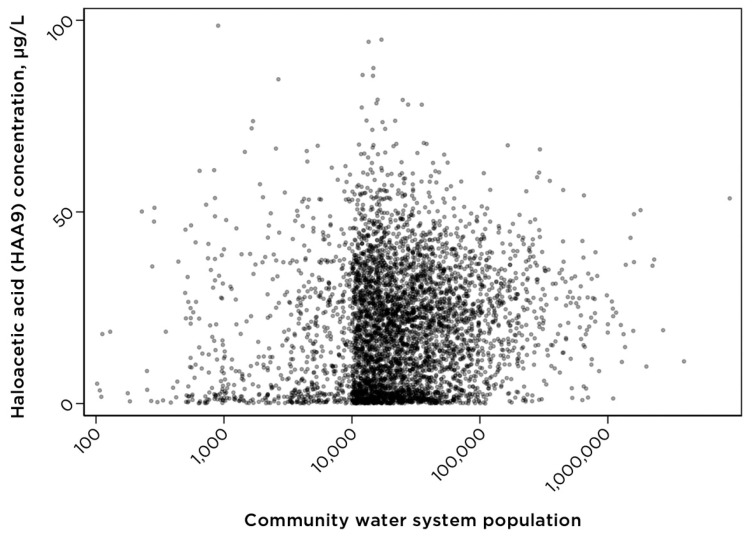
The concentration of 9 haloacetic acids for community water systems of different sizes. Community water systems serving less than 100 people (24 systems) and those with the concentration of HAA9 greater than 100 µg/L (10 systems) are not shown. Information on the population served by community water systems was obtained from the U.S. EPA Envirofacts database.

**Figure 2 ijerph-18-10401-f002:**
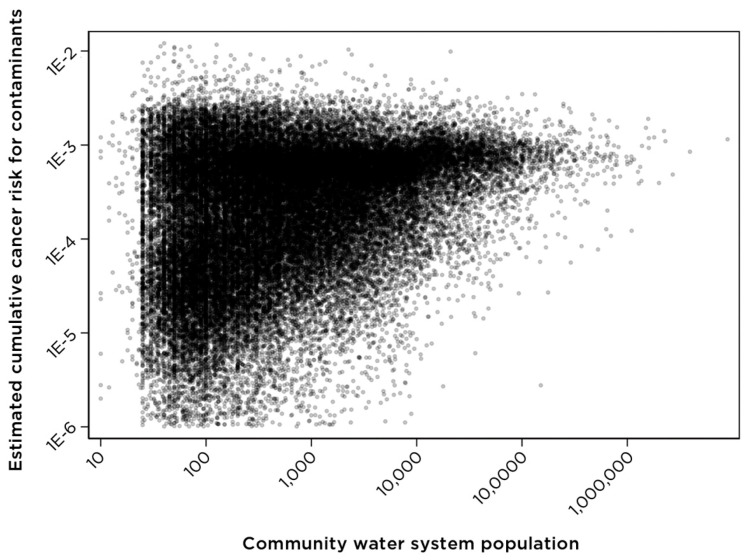
Estimated cumulative cancer risk for community water systems of different sizes. Information on the population served by community water systems was obtained from the U.S. EPA Envirofacts database.

**Figure 3 ijerph-18-10401-f003:**
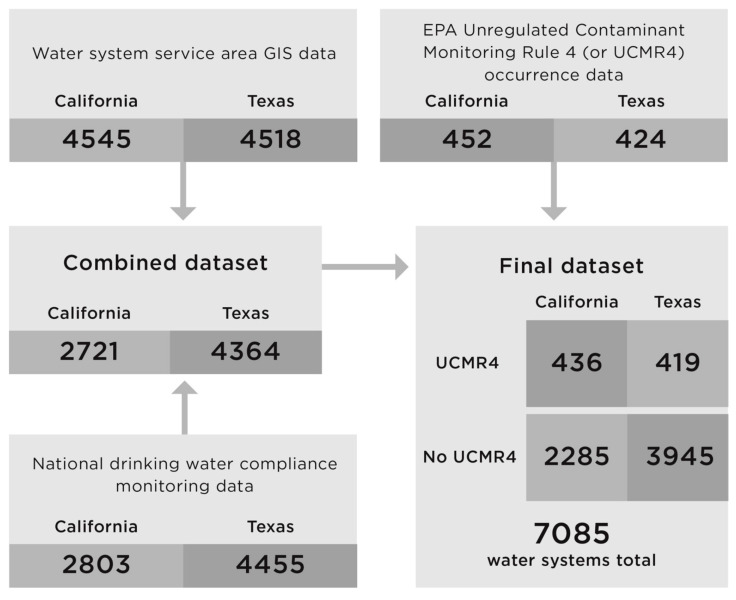
Contaminant occurrence and GIS service area data availability for community water systems in California and Texas. The flow chart shows the number of systems with specific data and the number of systems in the final combined data set.

**Figure 4 ijerph-18-10401-f004:**
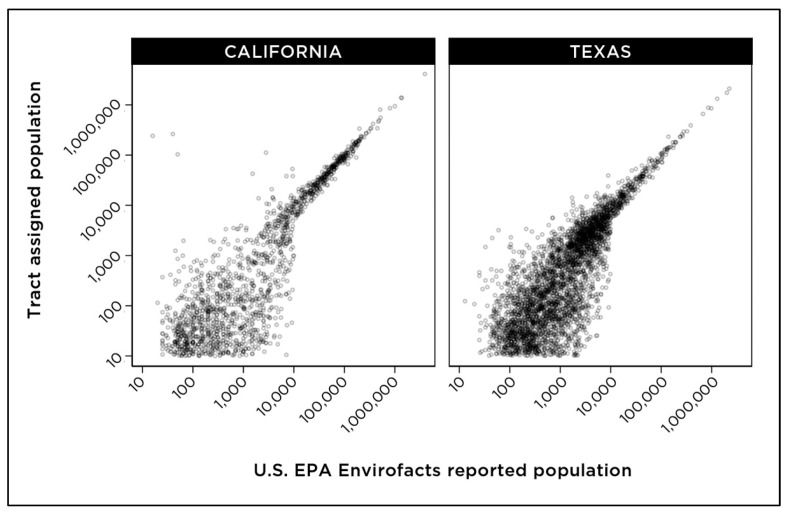
Population estimates calculated using census tract assignment compared with reported population statistics from the U.S. EPA Envirofacts database.

**Figure 5 ijerph-18-10401-f005:**
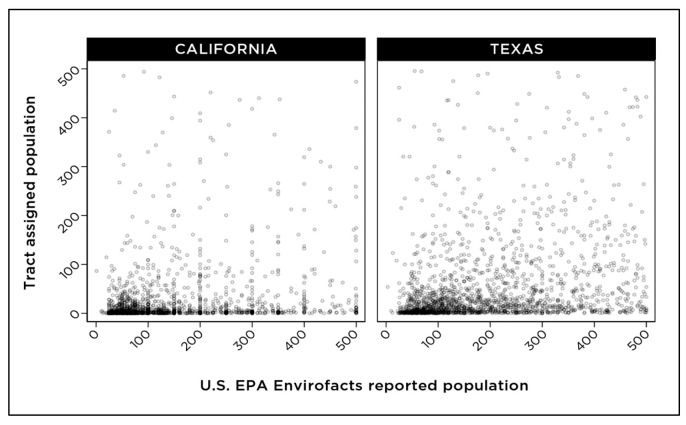
Tract-assigned population estimates for community water systems that, according to the U.S. EPA Envirofacts, serve populations of 500 people or less.

**Figure 6 ijerph-18-10401-f006:**
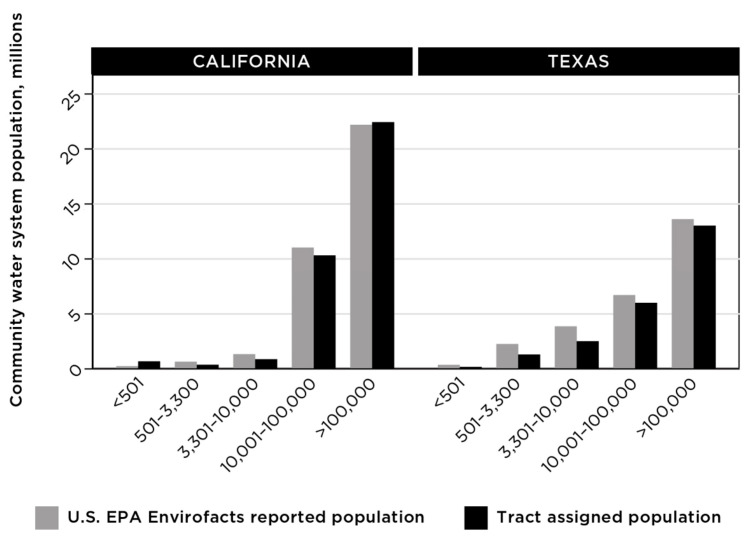
Comparison between the population served by community water systems according to data from the U.S. EPA Envirofacts database and the population calculated based on matching census tracts.

**Figure 7 ijerph-18-10401-f007:**
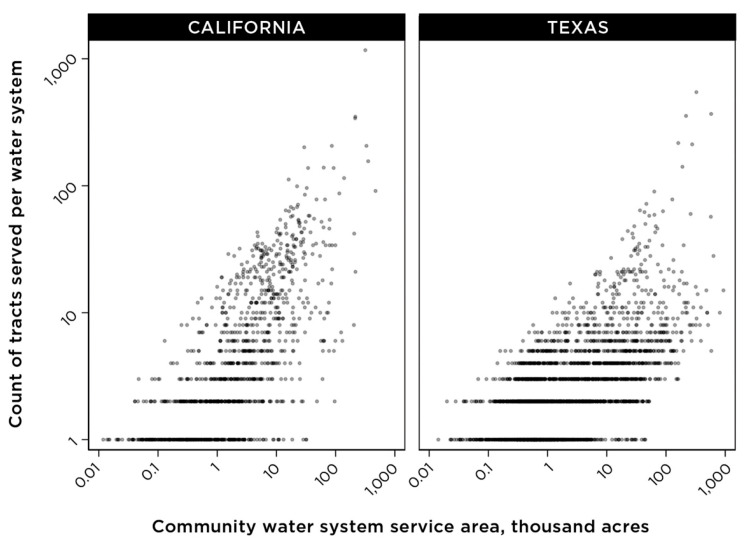
Number of unique census tracts that are partially or completely included in the service areas of community water systems in California and Texas. An acre, a unit of land measurement commonly used in the United States, corresponds to approximately 4047 m^2^.

**Figure 8 ijerph-18-10401-f008:**
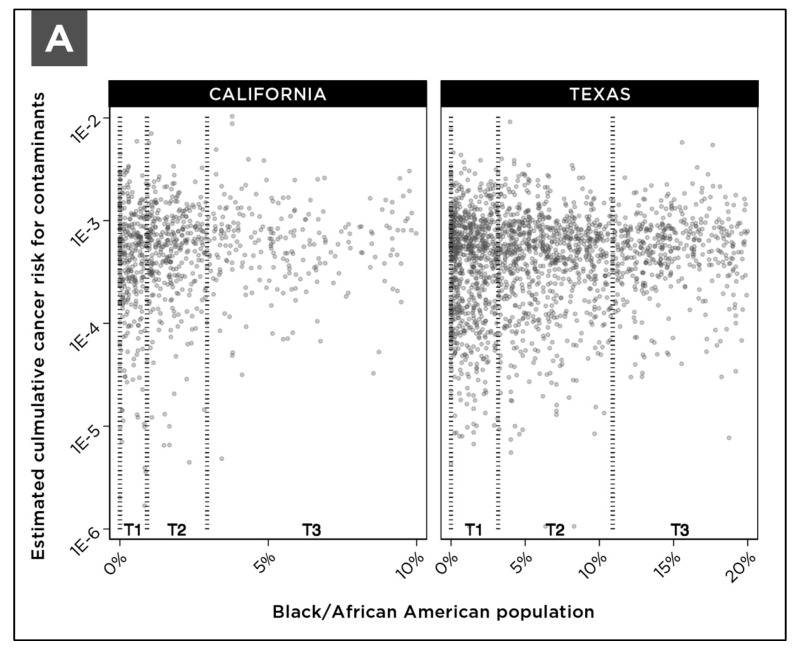
Distribution of cumulative cancer risk for community water systems in California and Texas divided into terciles according to the (**A**) percentage of the population that is Black/African American and (**B**) percentage of the population that is Hispanic/Latino within each system’s service area. The figure includes all systems in each state serving populations greater than 500 people. Terciles are indicated as T1, T2, and T3, and tercile boundaries are listed in Appendix A, Table A3.

**Figure 9 ijerph-18-10401-f009:**
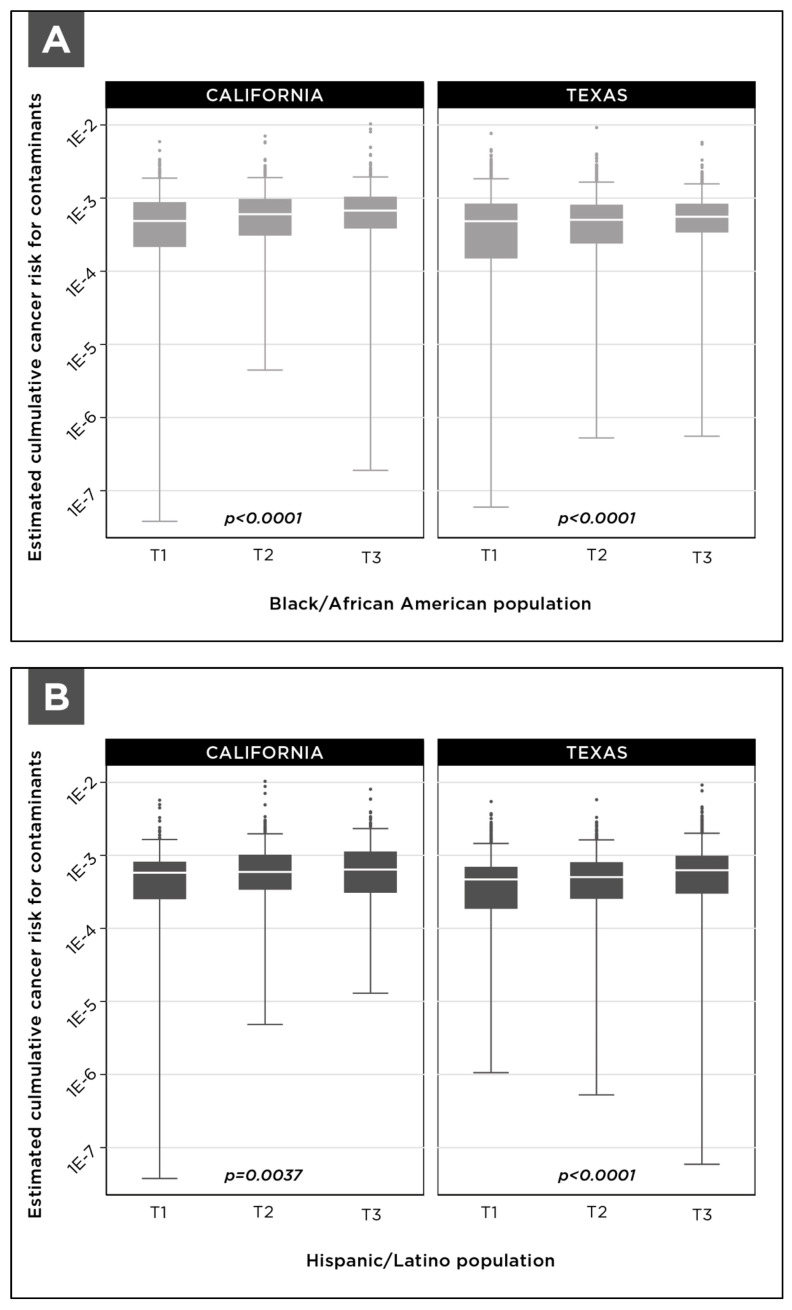
Cumulative cancer risk for community water systems in California and Texas grouped into terciles according to (**A**) the percentage of the population that is Black/African American and (**B**) the percentage of the population that is Hispanic/Latino within each system’s service area. Demographic percentages and total population for each tercile are listed in Appendix A, Table A3. The figure includes all systems in each state serving populations greater than 500 people.

**Table 1 ijerph-18-10401-t001:** Occurrence and average concentrations for contaminants included in this study.

Contaminant	Number of Systems with Contaminant Detections	Average Contaminant Concentration Across Water Systems of All Sizes	Population Served by Systems with Contaminant Detections, Millions ^1^	Population Weighted Average Contaminant Concentration
1,2-dibromo-3-chloropropane	213	0.1 ng/L	7.1	0.3 ng/L
1,2,3-trichloropropane	400	0.2 ng/L	12.5	0.4 ng/L
1,4-dioxane	1702	0.01 μg/L	89.2	0.1 μg/L
Arsenic	16,785	0.9 μg/L	127.1	0.6 μg/L
Benzene	190	0.9 ng/L	3.6	0.8 ng/L
Bromate	520	0.01 μg/L	29.2	0.1 μg/L
Carbon tetrachloride	434	1.9 ng/L	6.0	1.6 ng/L
Group of 5 haloacetic acid (HAA5)	31,648	8.1 μg/L	292.8	17 μg/L
Group of 9 haloacetic acids (HAA9)	4660	20 μg/L	249.8	24 μg/L
Hexavalent chromium	7821	0.1 μg/L	246.9	0.4 μg/L
Nitrate	35,555	0.9 mg/L ^2^	265.8	0.9 mg/L ^2^
Perchloroethylene	869	0.01 μg/L	28.8	0.02 μg/L
Radium-226 & -228	22,683	0.5 pCi/L	138.3	0.4 pCi/L
Strontium-90	87	0.0005 pCi/L	2.8	0.002 pCi/L
Group of 4 trihalomethanes (THM4) ^3^	35,937	16 μg/L	299.4	30 μg/L
Trichloroethylene	580	0.005 μg/L	22.7	0.02 μg/L
Tritium	84	0.3 pCi/L	1.8	1.0 pCi/L
Uranium	8226	0.6 pCi/L	74.4	0.5 pCi/L
Vinyl chloride	122	0.5 ng/L	2.1	1.8 ng/L

^1^ Information about the number of people served by community water systems was obtained from the U.S. EPA Envirofacts database (https://enviro.epa.gov/, accessed on 1 March 2020) and state drinking water programs. These population numbers represent an estimate, and the specific number of customers served by individual water systems in the data set may differ. ^2^ Nitrate concentration reported as nitrogen. For systems where only the combined concentration of nitrate and nitrite was reported, that value was included in the data array for nitrate. ^3^ Data from the U.S. EPA fourth Unregulated Contaminant Monitoring Rule. The UCMR4 data set analyzed here includes data for 4733 community water systems, serving a combined population of approximately 251 million people.

**Table 2 ijerph-18-10401-t002:** Cumulative cancer risk for community water systems in California grouped by system size.

Community Water System Size	Demographic Group Defined by the U.S. Census Bureau	Tercile Boundaries for Percentage of Population within a Demographic Group ^1^	Total Population in the Tercile, Millions (Rounded)	Median Cumulative Cancer Risk	Jonckheere-Terpstra Trend Test *p*-Value ^2^
Small (501–3300)	Black or African American	≤0.5% (T1)	0.21	4.3 × 10^−4^	0.1333
		0.5–2.2% (T2)	0.21	3.5 × 10^−4^	
		≥2.2% (T3)	0.23	5.9 × 10^−4^	
	Hispanic or Latino	≤14.3% (T1)	0.21	4.4 × 10^−4^	0.0431 *
		14.4–30.6% (T2)	0.22	4.9 × 10^−4^	
		≥30.9% (T3)	0.21	4.9 × 10^−4^	
Medium (3301–10,000)	Black or African American	≤0.7% (T1)	0.43	5.9 × 10^−4^	0.6883
		0.7–2.5% (T2)	0.46	5.8 × 10^−4^	
		≥2.6% (T3)	0.45	6.6 × 10^−4^	
	Hispanic or Latino	≤20.9% (T1)	0.42	5.8 × 10^−4^	0.4065
		20.9–43.2% (T2)	0.46	6.7 × 10^−4^	
		≥43.6% (T3)	0.45	6.0 × 10^−4^	
Large (10,001–100,000)	Black or African American	≤1.5% (T1)	3.1	6.7 × 10^−4^	0.1625
		1.5–3.7% (T2)	4.2	6.7 × 10^−4^	
		≥3.8% (T3)	3.7	7.8 × 10^−4^	
	Hispanic or Latino	≤23.0% (T1)	3.6	7.2 × 10^−4^	0.1183
		23.3–44.1% (T2)	3.8	8.0 × 10^−4^	
		≥44.5% (T3)	3.6	5.8 × 10^−4^	
Very Large (>100,000)	Black or African American	≤2.6% (T1)	4.6	5.6 × 10^−4^	0.0041 *
		2.7–6.4% (T2)	6.8	6.8 × 10^−4^	
		≥6.7% (T3)	10.8	7.6 × 10^−4^	
	Hispanic or Latino	≤30.0% (T1)	6.4	6.9 × 10^−4^	0.7324
		30.1–47.7% (T2)	10.2	6.9 × 10^−4^	
		≥48.0% (T3)	5.6	6.5 × 10^−4^	

^1^ The tercile boundaries are indicated with a single decimal digit and may appear to overlap due to rounding; actual tercile boundaries were defined with multiple decimal digits and do not overlap. ^2^ Trend test *p* values less than 0.05 are considered statistically significant and are marked with an asterisk (*).

**Table 3 ijerph-18-10401-t003:** Cumulative cancer risk for community water systems in Texas grouped by system size.

Community Water System Size	Demographic Group Defined by the U.S. Census Bureau	Tercile Boundaries for Percentage of Population within a Demographic Group ^1^	Total Population in the Tercile, Millions (Rounded)	Median Cumulative Cancer Risk	Jonckheere-Terpstra Trend Test *p*-Value ^2^
Small (500–3300)	Black or African American	≤2.3% (T1)	0.72	4.1 × 10^−4^	0.0005 *
		2.3–9.0% (T2)	0.75	4.3 × 10^−4^	
		≥9.0% (T3)	0.79	5.0 × 10^−4^	
	Hispanic or Latino	≤13.3% (T1)	0.74	4.2 × 10^−4^	<0.0001 *
		13.3–25.9% (T2)	0.76	4.2 × 10^−4^	
		≥25.9% (T3)	0.76	5.6 × 10^−4^	
Medium (3301–10,000)	Black or African American	≤5.2% (T1)	1.3	5.3 × 10^−4^	0.0598
		5.2–14.5% (T2)	1.3	4.8 × 10^−4^	
		≥14.6% (T3)	1.3	5.8 × 10^−4^	
	Hispanic or Latino	≤17.5% (T1)	1.3	5.2 × 10^−4^	0.0230 *
		17.5–33.6% (T2)	1.3	5.2 × 10^−4^	
		≥33.8% (T3)	1.4	5.5 × 10^−4^	
Large (10,001–100,000)	Black or African American	≤4.2% (T1)	2.2	8.0 × 10^−4^	0.9367
		4.3–12.6% (T2)	2.2	7.0 × 10^−4^	
		≥12.6% (T3)	2.3	8.0 × 10^−4^	
	Hispanic or Latino	≤21.5% (T1)	2.2	7.6 × 10^−4^	0.1826
		21.6–37.8% (T2)	2.1	7.3 × 10^−4^	
		≥37.8% (T3)	2.4	8.3 × 10^−4^	
Very Large (>100,000)	Black or African American	≤7.9% (T1)	5.4	9.1 × 10^−4^	0.1684
		8.0–13.2% (T2)	2.1	8.2 × 10^−4^	
		≥14.3% (T3)	6.2	7.4 × 10^−4^	
	Hispanic or Latino	≤26.1% (T1)	1.7	8.2 × 10^−4^	0.1270
		26.1–42.9% (T2)	5.0	7.9 × 10^−4^	
		≥43.0% (T3)	6.9	9.2 × 10^−4^	

^1^ The tercile boundaries are indicated with a single decimal digit and may appear to overlap due to rounding; actual tercile boundaries were defined with multiple decimal digits and do not overlap. ^2^ Trend test p values less than 0.05 are considered statistically significant and are marked with an asterisk (*).

## Data Availability

The drinking water data sets for each U.S. state can be obtained either via a direct data access portal provided by a state drinking water authority or via a records request. Links for the websites of government agencies from which data were obtained are listed in Appendix A Table A1.

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
