# Peer review of "Community-Level Analysis of Drinking Water Data Highlights the Importance of Drinking Water Metrics for the State, Federal Environmental Health Justice Priorities in the United States"

_ijerph, 2021, doi:10.3390/ijerph181910401_

Round 1

Reviewer 1 Report

It is really very important to consider the drinking water quality in advancing the environmental health justice priorities. The article present a framework for the joint consideration of community- level drinking water data and demographic data. And conducted a comprehensive analysis of the relationships and the impact of water quality on human health. It is a well organized article and with rich basic data.

My  suggestion is that The study would be more convincing if it could be supplemented with actual cancer rates in these typical communities (which may not be readily available). Risk assessment, after all, is only a theoretical possibility.

  More suggestions at the attachment.

Author Response

Responses to Reviewer # 1

It is really very important to consider the drinking water quality in advancing the environmental health justice priorities. The article present a framework for the joint consideration of community- level drinking water data and demographic data. And conducted a comprehensive analysis of the relationships and the impact of water quality on human health. It is a well organized article and with rich basic data.

We thank the reviewer for the insightful, detailed comments.

My  suggestion is that The study would be more convincing if it could be supplemented with actual cancer rates in these typical communities (which may not be readily available). Risk assessment, after all, is only a theoretical possibility.

We agree with the reviewer that supplementing our analysis with actual cancer rates would be helpful. However, such additional analysis is outside the scope and focus of this paper and would be suited for an additional future paper. This paper focuses on the development of a framework that can identify and describe patterns in drinking water quality on the community level which would help state and federal efforts in the U.S. to improve drinking water quality for all communities and to advance environmental health justice priorities. We also agree that the topic of cancer disparities needs to be mentioned. Hence, the revised manuscript cites data from the National Cancer Institute's Surveillance, Epidemiology, and End Results (SEER) database about cancer disparities in the U.S. and this appears on lines 445-447 – “According to the U.S. National Cancer Institute's Surveillance, Epidemiology, and End Results (SEER) database, cancer mortality rates are higher among the Black/African American population than other groups.”

Although the paper is not methodologically or academically innovative. Given that drinking water does have a direct impact on human health, the manuscript provides some trend results for drinking water safety risks based on a comprehensive analysis of a large number of published drinking water quality data. The reliability of these results mainly comes from the rationality of the evaluation method of water quality safety risk. Cumulative lifetime cancer risk index R was used to evaluate the safety risk of water quality. R is cumulative lifetime cancer risk per million people on a system level due to all drinking water contaminants. The correctness of cumulative cancer risk index R depends on the scientificity and rationality of the cancer risk benchmarks (the value of Bi ). Objectively speaking, cancer incidence is a medical indicator, influenced by a variety of factors. Whether these artificial assigned benchmarks are correct still requires actual evidence to test. That's why I mentioned in my last review that it would be good to provide a recommendation on the actual incidence of cancer in the typical California and Texas area studied. If the actual cancer incidence results are consistent with the risk assessment, the results of this paper could serve as a reference for future government investment in community health and public policies-making to advance environmental health justice priorities.

We agree that this investigation is important within the environmental health field and acknowledge the reviewer’s request for additional information. As noted above, our article focuses on the development of a framework to analyze environmental health disparities using public drinking water data and the latest cancer risk estimates. More so, the goal of this study is to identify trends that can ultimately inform policy and catalyze interventions that encourage environmental equity using currently available data. The cumulative cancer risk estimated in this study integrates information about the overall risk from co-occurring carcinogenic contaminants in drinking water, and this is based on the additive risk of the one-in-a-million cancer risk benchmarks established by state agencies. The current study applies this cumulative cancer risk framework that was described in several previously published studies that presented this additive model of established cancer risk benchmarks based on epidemiological and toxicological data [1-4]. We believe the cancer risk framework can be refined, and analysis of cancer risk estimates and incidence could be a topic of future research. As noted above, we included more information about cancer disparities in addition to our risk estimates for these two states.

Reviewer 2 Report

I Have some questios: 

Table 1, is presented the concentrations of the contaminantes. However, the numbers described in the other colluns are about other information, not about the concentrataion of the contaminants. I suggest to revise the table. 

Along the manuscript, authors described the cancer risk. However, I would like to know if the contaminant concentration escribed are above the regulations recomendations. Once you do not discuss the concentration, it could be a missunderstand by the readers, and a superestimation of the risk. It is also important to realize that the presence of contaminants is not the only fact related to cancer. The genetic condition is also important to highlight. 

3.3 The authors wrote "“Black or African American” and “Hispanic or Latino origin". Is it the correct word? In my country we can not use the word "black" for people. 

Author Response

Responses to Reviewer # 2

Table 1, is presented the concentrations of the contaminantes. However, the numbers described in the other colluns are about other information, not about the concentrataion of the contaminants. I suggest to revise the table. 

Thank you for the comment. We have revised Table 1 to include the average concentrations of contaminants as suggested by the reviewer.

Along the manuscript, authors described the cancer risk. However, I would like to know if the contaminant concentration described are above the regulations recomendations. Once you do not discuss the concentration, it could be a missunderstand by the readers, and a superestimation of the risk.

Thank you for the comments. We included the list of regulatory standards for contaminants evaluated in this study and this appears in Appendix A, Table A2. Also, a discussion comparing values presented in this study to the regulatory standards was added to the result section and this appears on lines 163-170: The U.S. EPA published maximum contaminant levels, or MCL, which are the legal standards for community water systems established under the U.S. Safe Drinking Water Act (Appendix A Table A2). The average concentrations of contaminants (Table 1) are lower than the legal limits established by the U.S. EPA (Table A2). However, it is important to recognize that numerous community water systems in the U.S., serving millions of people, are not able to comply with the existing standards and have average contaminant concentrations that are close to or violate the existing legal standards.

It is also important to realize that the presence of contaminants is not the only fact related to cancer. The genetic condition is also important to highlight. 

Thank you for the comment. The authors agree that exposure to environmental contaminants is not the only influence on cancer risk. We edited the text to highlight other factors including genetic conditions that could influence cancer risk in our discussion. This can be found in lines 497-501 “Multiple factors such as genetic background and individual life circumstances can influence cancer risk. The assessment of cumulative exposures as well as other risk factors should become a cornerstone for future policies to improve environmental health in all communities, especially communities that historically experienced discrimination and may still face a disproportionate burden of pollution to date.”

3.3 The authors wrote "“Black or African American” and “Hispanic or Latino origin". Is it the correct word? In my country we can not use the word "black" for people.

Thank you for the comment. The authors used the phrasing “Black or African American and Hispanic or Latino” as used by the U.S Census Bureau in their surveys. This has been clarified in the manuscript and now appears at lines 302-303 “The demographic terms used in our study are the terms used by the U.S. Census.”

Reviewer 3 Report

In considering the purpose of this research, it seems that the selection of research methodology and scope of analysis, the process of deriving alternatives, and the specificity are sufficiently systematic as an academic article. However, in theoretical composition, this research requires supplementation since the review of preceding studies is somewhat limited when quotations are taken into consideration. In addition, in order to explain the policy alternatives logically, separate drawing and presentation of implications regarding case analyses seem to be more useful. This aspect should be reflected in the modification of this article.

Author Response

Responses to Reviewer # 3

In considering the purpose of this research, it seems that the selection of research methodology and scope of analysis, the process of deriving alternatives, and the specificity are sufficiently systematic as an academic article.

We thank the reviewer for the comment.

However, in theoretical composition, this research requires supplementation since the review of preceding studies is somewhat limited when quotations are taken into consideration.

We have expanded the discussion section to include review of previous studies on drinking water quality and this can be found at lines 417-434 “The results of our analysis are consistent with published literature finding social disparities in drinking water quality, such as the findings of a study conducted in California’s San Joaquin Valley that reported higher arsenic concentrations in drinking water and higher odds of having a violation of drinking water standards serving predominantly socio-economically disadvantaged communities [1]; the findings of a U.S.-wide study that reported a significant association between the percent of Hispanic residents served by a community water system and average concentrations of nitrate in drinking water [2]; and the findings of a U.S.-wide study conducted by the Natural Resources Defense Council, Environmental Justice Health Alliance for Chemical Policy Reform and Coming Clean which found that water systems serving communities of color had a higher rate of violations of national drinking water standards [3]. The findings of our study, together with prior research, highlight how communities of color disproportionately face worse drinking water quality, which can increase their risk for adverse health impacts such as the elevated risk of cancer. These disparities in environmental contaminant exposure are further aggravated by the fact that communities and populations of color in the United States continue to experience greater health inequalities [4] in general. They also have less access to adequate health care compared to the other populations [5].”

In addition, in order to explain the policy alternatives logically, separate drawing and presentation of implications regarding case analyses seem to be more useful. This aspect should be reflected in the modification of this article.

We thank the reviewer for the insightful, detailed comments. We applied our utmost efforts to update the study in accordance with the reviewer’s suggestions and included policy implications and recommendations throughout the discussion section and conclusion. For example, we have added this statement at lines 436-439 “Including drinking water data metrics in state and federal environmental justice analyses used for policy decisions and other actions would be an important step to address the identified disparity while also offering better health protections for all communities.” At lines 501-503 “Implementing the data and methods presented here into screening tools can supplement and support community-level knowledge in facing environmental challenges.”

  1. Balazs, C.L., et al., Environmental justice implications of arsenic contamination in California's San Joaquin Valley: a cross-sectional, cluster-design examining exposure and compliance in community drinking water systems. Environ Health, 2012. 11: p. 84.
  2. Schaider, L.A., et al., Environmental justice and drinking water quality: are there socioeconomic disparities in nitrate levels in US drinking water? Environmental Health, 2019. 18(1): p. 1-15.
  3. Fedinick, K.P., S. Taylor, and M. Roberts. Watered Down Justice. Natural Resource Defence Council 2019 March 27,2020 August 11, 2021]; Available from: https://www.nrdc.org/resources/watered-down-justice.
  4. Gehlert, S., D. Hudson, and T. Sacks, A Critical Theoretical Approach to Cancer Disparities: Breast Cancer and the Social Determinants of Health. Front Public Health, 2021. 9: p. 674736.
  5. LaVeist, T.A., Minority populations and health: An introduction to health disparities in the United States. Vol. 4. 2005: John Wiley & Sons.

Round 2

Reviewer 2 Report

The article can be accepted.